**PLOS** NEGLECTED TROPICAL DISEASES

# Chikungunya virus (CHIKV) seroprevalence in the South Pacific populations of the Cook Islands and Vanuatu with associated environmental and social factors

**Charlotte E. B. Saretzki**[1]*, **Gerhard Dobler**[2], **Elisabeth Iro**[3], **Yin May**[4], **Douglas Tou**[5], **Eteta Lockington**[6], **Michael Ala**[7], **Nicole Heussen**[8,9], **Bruno S. J. Phiri**[10], **Thomas Küpper**[1]

**1** Institute for Occupational, Social and Environmental Medicine, RWTH Aachen Technical University, Aachen/ Germany, **2** Bundeswehr Institute of Microbiology, Munich/ Germany, **3** Cook Islands Ministry of Health, Rarotonga/ Cook Islands, **4** Cook Islands Ministry of Health, Rarotonga Hospital, Rarotonga/ Cook Islands, **5** Cook Islands Ministry of Health, Rarotonga Laboratory, Rarotonga/ Cook Islands, **6** Cook Islands Ministry of Health, Aitutaki Laboratory, Aitutaki/ Cook Islands, **7** Northern Provincial Hospital Laboratory, Espiritu Santo/ Vanuatu, **8** Department of Medical Statistics, RWTH Aachen Technical University, Aachen/ Germany, **9** Center of Biostatistics and Epidemiology, Medical School, Sigmund Freud University, Vienna/ Austria, **10** Central Veterinary Research Institute (CVRI), Ministry of Fisheries and Livestock, Lusaka/ Zambia

* charlotte.saretzki@rwth-aachen.de

**Data Availability Statement:** All data are in the manuscript and/or supporting information files.

## Abstract

### Background

Arthropod-borne diseases pose a significant and increasing risk to global health. Given its rapid dissemination, causing large-scale outbreaks with severe human infections and economic loss, the Chikungunya virus (CHIKV) is one of the most important arboviruses worldwide. Despite its significance, the real global impact of CHIKV remains underestimated as outbreak data are often incomplete and based solely on syndromic surveillance. During 2011–2016, the South Pacific Region was severely affected by several CHIKV-epidemics, yet the area is still underrepresented in arboviral research.

### Methods

465 outpatient serum samples collected between 08/2016 and 04/2017 on three islands of the island states Vanuatu (Espiritu Santo) and the Cook Islands (Rarotonga, Aitutaki) were tested for anti-CHIKV specific antibodies using Enzyme-linked immunosorbent Assays.

### Results

A total of 30% (Cook Islands) and 8% (Vanuatu) of specimens were found positive for anti-CHIKV specific antibodies with major variations in national and intranational immunity levels. Seroprevalence throughout all age groups was relatively constant. Four potential outbreak-protective factors were identified by comparing the different study settings: presence of Ae. albopictus (in absence of ECSA E1-A226V-mutation CHIKV), as well as low levels of human population densities, residents' travel activity and tourism.

**Funding:** The study was supported by Aachen Dental and Medical Expeditions (ADEMED e.V. https://www.ademed.de/), a non-profit society to support research in travel medicine grant number 180627 to CS. (funding: financial) Also by the Bundeswehr Institute of Microbiology, Munich to CS. (funding: materials and use of their laboratory [no grant / grant number]). The funders had no role in study design, data collection and analysis, decision to publish, or preparation of the manuscript.

**Competing interests:** The authors declare that there are no financial, personal, or professional interests that could be construed to have influenced the work.

## Conclusion

This is the first seroprevalence study focussing on an arboviral disease in the Cook Islands and Vanuatu. It highlights the impact of the 2014/2015 CHIKV epidemic on the Cook Islands population and shows that a notable part of the Vanuatu test population was exposed to CHIKV although no outbreaks were reported. Our findings supplement the knowledge concerning CHIKV epidemics in the South Pacific Region and contribute to a better understanding of virus dissemination, including outbreak modifying factors. This study may support preventive and rapid response measures in affected areas, travel-related risk assessment and infection identification in returning travellers.

## Trial registration

ClinicalTrials.gov Aachen: 051/16_09/05/2016 Cook Islands Ref.: #16-16 Vanuatu Ref.: MOH/DG 10/1/1-GKT/lr.

## Author summary

Arboviral infections pose an increasing risk to global health. As most of them predominantly circulate in (sub-)tropical regions comprised of low-resource countries and affect already vulnerable populations, these diseases have an immense impact on the socio-economic sector. Worldwide, the Chikungunya virus (CHIKV) represents one of the most important arboviruses. Yet, despite its significance, reliable data to understand many populations' real disease burden is still lacking. One example is the South Pacific Region (SPR), which has been severely affected by numerous CHIKV outbreaks since 2011. Epidemiological data is indispensable for the implementation of strategies for optimal allocation of limited resources, efficient early intervention, and vector control as well as to understand the virus' geographical distribution and its contribution to global morbidity. We therefore conducted a CHIKV seroprevalence study in the local populations of two Pacific Island states–the Cook Islands and Vanuatu. Our results show that about 30% of the local population has been affected during the 2014/2015 CHIKV epidemic on the Cook Islands. Although no outbreak was ever officially reported in Vanuatu, a notable part of the respective test population was also proven to have been exposed to CHIKV. By comparing our two study settings we further identified several environmental and social factors with potential outbreak modifying effect.

## 1. Introduction

Arboviral infections pose a significant and increasing challenge to global health [1]. Despite their very significant effect on public health and society, including economy and social structures, exact information concerning the real local and global burden of disease is often lacking [2]. Amongst all arboviruses, today an entity that for a long time received little notice adds to the challenges many regions' health systems must face: The Chikungunya Virus (CHIKV)—an enveloped, RNA Alphavirus belonging to the family of Togaviridae [3]. CHIKV appears in the three distinct phylogroups West African (WA), East/Central/South African (ESCA), and Asian [4] and is transmitted through bites of *Aedes* spp. mosquitoes, mainly *Aedes aegypti* and *Aedes albopictus* [5]. Starting in 2005, large-scale outbreaks in the Indian and the Pacific

Ocean area as well as on the American continent, not only highlighted the virus' potential for rapid dissemination causing epidemics of hitherto unprecedented extent, but also revealed severe complications linked to human infection including crippling and persisting arthralgias, neurological disorders and cardiovascular manifestations [6,7,8].

One of the areas severely affected by CHIKV, but still largely missing from the epidemiological map, is the South Pacific Region (SPR) [8,9,10].The area has historically been free of CHIKV [11], but since its introduction in 2011 the virus has disseminated throughout the region causing numerous outbreaks (Fig 1). The SPR represents a very distinct geographical setting: It is characterized by a vast area of open ocean with thousands of islands scattered in between which comprise the 22 Pacific Island Countries and Territories (PICTs) [12]. These can be subdivided into the three subregions Melanesia, Micronesia, and Polynesia and are home to approximately 11.4 million pacific islanders [13]. Many PICTs fall into the United

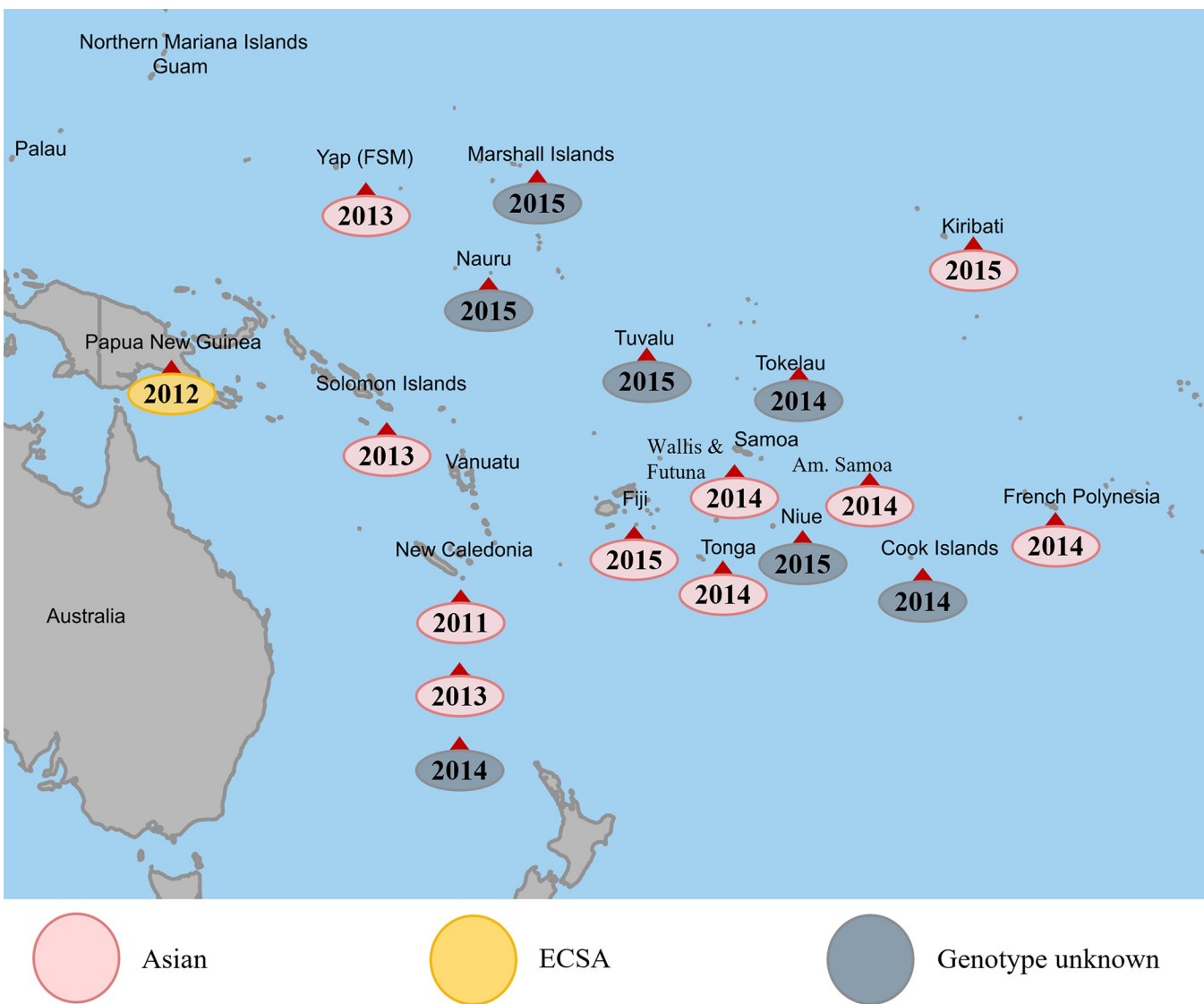

**Fig 1. History of CHIKV outbreaks in the SPR, including genotype if known.** Source: [10,13,17,18,19,20,21,22,23,24,25,26], base layer made with Natural Earth (naturalearthdata.com).

Nations Development Programme's categories *developing* or *least developed countries* and are ranked among the countries most at risk from natural disasters [12]. The combination of its archipelagic geography, tropical climate, presence of potent vectors, naive populations, development status and highly frequent population flows including millions of tourists each year, not only makes the region particularly vulnerable to arboviral epidemics but also provides nearly ideal outbreak conditions [9,10,13,14,15,16,17]. Despite a high local burden of disease, the SPR is therefore assumed to have greatly contributed to the global spread of DENV, ZIKV and CHIKV [9,10] and a wider knowledge about its arboviral situation is of international interest.

## 2. Materials and methods

### Ethics statement

The study was conducted in accordance with the Ethics Committee of the Medical Faculty of the Rheinisch-Westfaelische Technische Hochschule (RWTH) Aachen University (051/16_09/05/2016) and with the local authorities of the Cook Islands (Ref.: #16–16) and Vanuatu (Ref.: MOH/DG 10/1/1-GKT/lr).

To evaluate the seroprevalence of CHIKV the study has been conducted in the local populations and permanent inhabitants of the Cook Islands, which have been affected by an outbreak in 2014/2015 and Vanuatu, which had never reported such an outbreak. In choosing these two study locations we were able to compare seroprevalence rates of a Melanesian country in the western part of the SPR to those of a Polynesian country in the east. This is of interest as the two regions show considerable differences in terms of population mobility. In addition, several environmental variables possibly influencing vector spread were assessed.

Serum samples were collected in hospital laboratories of the island states Vanuatu (outer island Espiritu Santo) during August 2016 –January 2017 and the Cook Islands (main island Rarotonga and outer island Aitutaki) during January 2017 –April 2017. On each island, there was only one hospital laboratory. Residents requiring a blood test within the hospitals' normal diagnostic routine were invited to participate in the study. Tourists and short-term visitors were excluded. After obtaining written informed consent, basic epidemiological information was collected via questionnaires and interviews. In total, 465 samples (197 from Espiritu Santo, 208 from Rarotonga and 60 from Aitutaki) were analysed for anti-CHIKV specific antibodies using a recombinant-antigen-based indirect Enzyme-linked immunosorbent Assay (ELISA) (EUROIMMUN (EI 293a-9601 G); sensitivity: 95.4%; specificity: 98.6%). Test results were defined as "positive", "negative" or, if in-between the threshold values, "equivocal".

Associations between ELISA test results and gender as well as between seroprevalence levels and test collective were performed by means of $Chi^2$-Test. To test for associations between ELISA test results and age (age groups 'adults' (>15 years) and 'children' (0–15 years)), the Fisher-Freeman-Halton Exact Test was used due to smaller test collectives and percentages of >20% of expected frequencies below 5. For all comparisons the significance level was set to 5%; due to the explorative nature of the investigation no adjustment to the significance level was made. Results were reported as percentage and two-sided p-values. For estimation and comparison of community immunity-levels, raw data were directly standardized by age (10-year age groups; excluding the age group of 0–9 years due to low numbers) and gender according to the standard populations "total resident population" (Cook Islands) and "total population living in private households" (Vanuatu) as published in the Cook Islands Census of Population and Dwellings, 2011 [27] and the Vanuatu Post-Tropical Cyclone Pam Mini-Census, 2016 [28] respectively (further referred to as "resident population" or "residents"). Estimates of immunity in the different collectives were accompanied by 95% confidence limits (Cl) and p-values.

**Table 1. Environmental and social data of epidemiological relevance in the study sites.**

| Location | Cook Islands (total) | Rarotonga | Aitutaki | Vanuatu (total) | Espiritu Santo | Reference |
|---|---|---|---|---|---|---|
| **Pacific Subregion** | Polynesia | Polynesia | Polynesia | Melanesia | Melanesia | 15 |
| **Number of main islands** | 15 | | | 83 | | [29,30] |
| **Total size [km$^2$]** | 1.8 Mio | | | 360,000 | | [29,30] |
| **Landmass [km$^2$]** | 240 | 67.1 | 18.3 | 12,281 | 3,677 | [27,29,30,31] |
| **Total resident population** | 14,974 | 10,572 | 1,771 | 266,555 | 47,899 | [27,28] |
| **Pop. density (residents) [1/km$^2$]** | 62 | 158 | 97 | 22 | 13 | |
| **Aedes mosquitoes present** | *aegypti, polynesiensis* | *aegypti, polynesiensis* | *aegypti, polynesiensis* | *aegypti, albopictus, hebrideus* | *aegypti, albopictus, hebrideus* | [17,32,33] |
| **Temperature [˚C] (warmest month)** | 26.1 (Feb.) | | | 26.0 (Jan.) | | [34] |
| **Temperature [˚C] (coldest month)** | 21.7 (Aug.) | | | 23.5 (Jul.) | | [34] |
| **Highest precipitation [mm] (month)** | 241 (Jan.) | | | 375 (Feb.) | | [35] |
| **Lowest precipitation [mm] (month)** | 95 (Oct.) | | | 137 (Sep.) | | [35] |
| **Avg. annual int. arrivals by air (residents & visitors, 2013–2016)** | 140,302 | 140,302 | No int. airport | 126,901 | 4,985 | [36,37] |
| **Avg. annual int. arrivals by air (visitors only, 2013–2016)** | 128,459 | 128,459 | No int. airport | 100,997 | 4,541 | [36,37] |
| **Avg. annual int. arrivals by air (residents only, 2013–2016)** | 11,843 | 11,843 | No int. airport | 25,904 | 444 | [36,37] |
| **Avg. annual int. arrivals by air per resident (residents only, 2013–2016)** | 0.791 | 1.12* | No int. airport | 0.097 | 0.009 | |
| **Avg. annual int. arrivals by air per resident (visitors only, 2013–2016)** | 8.58 | 12.15 | No int. airport | 0.38 | 0.1 | |

*Data not representative for Rarotonga residents as all residents travel via Rarotonga

Information concerning potentially outbreak-modifying environmental and social factors was obtained from local authorities as well as from the DWD Climate Data Center. Data regarding the Cook Islands CHIKV outbreak derive from the archives of the Pacific Public Health Surveillance Network. All analyses were performed with Microsoft Excel Office 365 and IBM SPSS Statistics 21. The base layer of the map was made with Natural Earth, free vector and raster map data @ naturalearthdata.com.

## 3. Results

Important epidemiological, environmental, and social data of the different study sites are displayed in Table 1.

According to the specimen origin, we mainly considered 2 different test populations in our analysis: The Cook Islands and the Vanuatu collective. The Cook Islands collective can be further divided into two subgroups corresponding to the different islands of origin, Rarotonga and Aitutaki (Table 2).

**Table 2. Test population structure–gender distribution and median age.**

| Location | Both sexes [No. samples] | Male [No. samples] | Male [%] | Female [No. samples] | Female [%] | Median age [years] |
|---|---|---|---|---|---|---|
| Cook Islands | 268 | 122 | 45.5 | 146 | 54.5 | 49 |
| Rarotonga | 208 | 83 | 39.9 | 125 | 60.1 | 47 |
| Aitutaki | 60 | 39 | 65 | 21 | 35 | 53.5 |
| Vanuatu (Espiritu Santo) | 197 | 80 | 40.6 | 117 | 59.4 | 36 |
| Total | 465 | 202 | 43.4 | 263 | 56.6 | — |

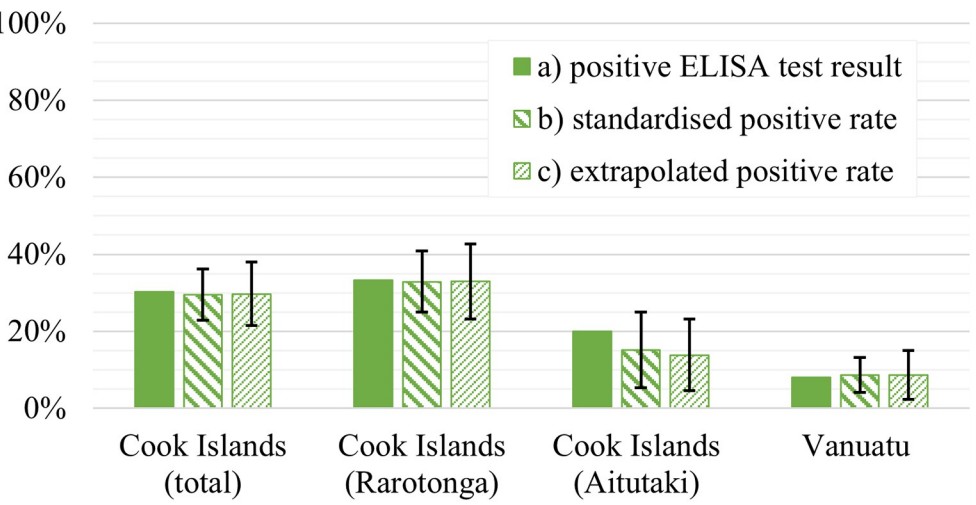

**Fig 2.** a) positive ELISA test results. b) seroprevalence rates standardized by gender and age. c) seroprevalence rates extrapolated to the total resident population. Black whiskers show 95%CI relative to standardized and extrapolated rate.

Anti-CHIKV specific antibodies could be found in 30.2% (81 individuals) of all Cook Islands specimens and in 8.1% (16 individuals) of the Vanuatu test collective. Further subdividing the Cook Islands test group, a larger percentage of the Rarotonga specimens was identified as positive (33.2%; 69 individuals), compared to the serum samples from Aitutaki (20%, 12 individuals) (Fig 2).

Using the Chi$^2$- and the Fisher-Freeman-Halton Exact Test respectively, we could show that in our two main test collectives, there was neither a significant association of ELISA test results and gender (Chi$^2$-Test p-values: >0.05) nor significant differences between the age groups 'adults' (>15 years) and 'children' (0–15 years) (Fisher-Freeman-Halton Exact Test, p-values: >0.05). Instead, seroprevalence levels were relatively constant throughout age groups without variations of statistical significance (Fisher-Freeman-Halton Exact Test, p-values: >0.05) (Fig 3).

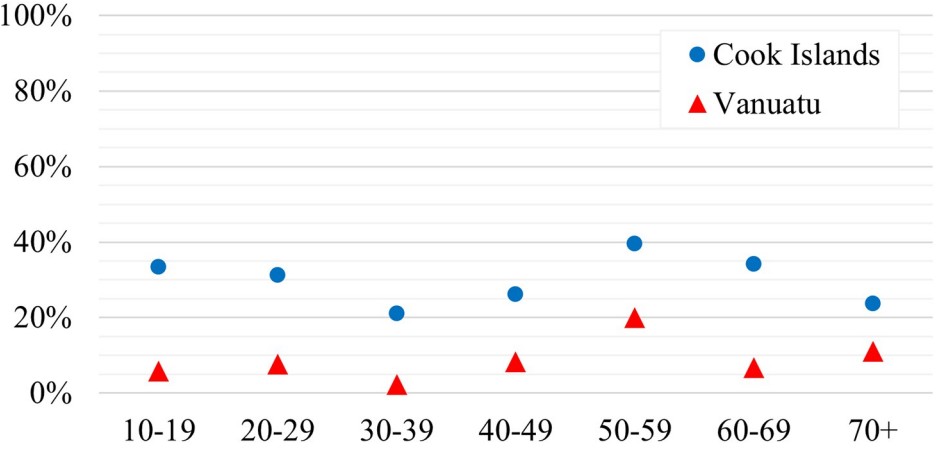

**Fig 3. Positive ELISA test results by 10 years age groups (excluding age group 0–9 years).**

To allow a direct comparison between our test collectives, data were standardized by age and gender (Fig 2) revealing significant differences in immunity rates for CHIKV between the Cook Islands (29.5%; 95%CI: 22.9–36.1) and the Vanuatu test collectives (8.6%; 95%CI: 4.1–13.2) (Chi$^2$-Test p-value: <0.05). Differences in seroprevalence levels were also significant between our two Cook Islands sub-populations from Rarotonga (32.9%; 95%CI: 24.9–40.8) and Aitutaki (15.1%; 95%CI: 5.3–25.0) (Chi$^2$-Test p-value: <0.05). Extrapolated to the total population of both island states, calculated immunity levels sum up to 29.7% (95%CI: 21.4–38.0) in the Cook Islands' total population and to 8.6% (95%CI: 2.2–15.0) in Vanuatu showing significant difference (Chi$^2$-Test p-value: <0.05) (Fig 2). In the two Cook Islands sub-populations, extrapolated rates amount to 32.9% (Rarotonga) (95%CI: 23.2–42.6) and 13.8% (Aitutaki) (95%CI: 4.5–23.1) as well showing a significant difference (Chi$^2$-Test p-value: <0.05).

## 4. Discussion

Subsequent to the 2014/2015 Cook Islands CHIKV epidemic, 30% of our test population showed evidence for prior CHIKV infection. Significant differences between both subgroups depict the effects of a scattered archipelagic geography on arboviral dissemination and are in line with surveillance data: During the outbreak hundreds of clinical cases were reported on the main island Rarotonga, but only five other islands, including Aitutaki, registered any infections and except for one, all index cases had travel history to Rarotonga [38]. The history of the Cook Islands CHIKV epidemic as reported by local surveillance systems is displayed in Fig 4. Between 10/2014 and 08/2015 there were 18 confirmed and 782 clinically diagnosed cases [13,39]. Considering our extrapolated results this corresponds to a calculated case-detection-rate of 18% for this specific outbreak.

The absence of significant differences between seroprevalence rates in children and adults as well as relatively little variation in immunity rates among 10-years age groups, suggest the primal epidemic and self-limiting occurrence of CHIKV in the Cook Islands, again consistent with surveillance data [26]. However, surveillance data are especially useful for outbreak detection but fail to depict the real burden of disease in a population [2]. They often remain incomplete, sporadic or delayed and are known to underestimate the dimensions of an epidemic [2,13]—an effect once again demonstrated by our calculated case-detection-rate of 18% for the Cook Islands CHIKV outbreak.

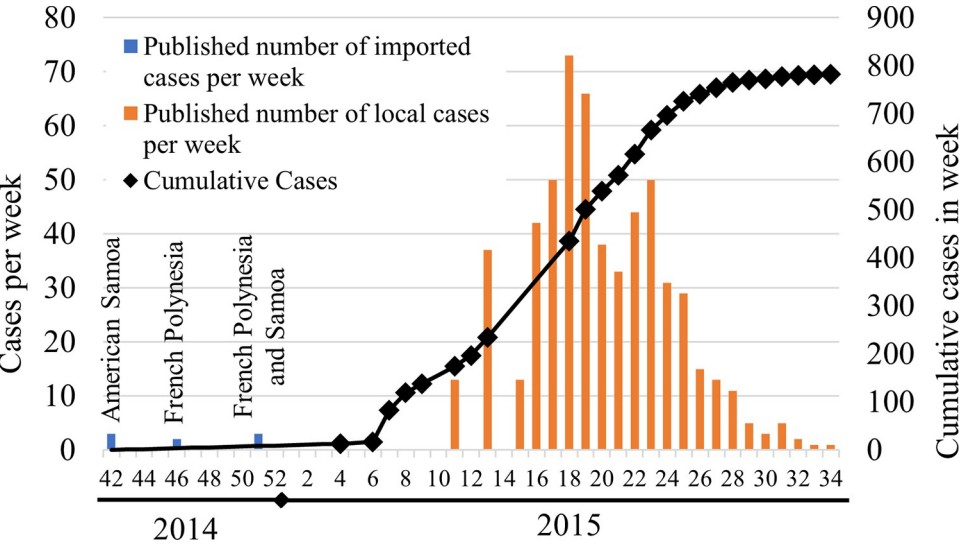

**Fig 4. 2014/2015 Cook Islands CHIKV outbreak as reported by local syndromic surveillance systems.** Source: [26]

Although Vanuatu never reported CHIKV circulation, our data provide evidence that 8% of the test collective had been previously exposed to the virus. Seropositivity was most likely acquired during the phase of multiple outbreaks in the SPR, and infections may have occurred either abroad and travel-related or during very localized outbreaks that never reached epidemic extent.

Explanatory approaches for the differences in seroprevalence levels observed in this study require the investigation of environmental and social driving factors of arboviral spread. For this purpose, we compared our study geographical settings in terms of vectors, climate, population density, mobility of the residential population and tourism (Table 1). Both settings host the competent vector *Ae. aegypti*. In terms of secondary vectors, the Cook Islands are home to *Ae. polynesiensis* while *Ae. albopictus* is prevalent in Vanuatu [17,32]. Another species of the genus *Aedes* present in Vanuatu is *Ae. hebrideus* [32], but since data concerning CHIKV transmission for this mosquito are scarce it is not further discussed within this work. Regarding the epidemiological effect of the two secondary vectors *Ae. albopictus* and *Ae. polynesiensis*, there are, however, some major differences: *Ae. polynesiensis* is particularly well-suited to PICT ecosystems as it uses artificial as well as a variety of natural breeding grounds including highly specific biotopes such as land crab burrows and fallen coconuts [32]. It therefore occurs in multiple environments and is very likely to act synergistically to *Ae. aegypti* in virus dissemination [32,40,41]. In contrast, *Ae. albopictus* has occasionally been shown to replace the main vector, *Ae. aegypti* [42]. Further, CHIKV transmission by *Ae. albopictus* mosquitoes is directly associated with an E1-A226V mutation in the ECSA-CHIKV envelope protein [43], while the Asian CHIKV lineage in general is limited in its ability to adapt to this vector [44]. *Ae. albopictus*' status as secondary vector in Vanuatu is thus CHIKV genotype-dependent, and due to its tendency to replace *Ae. aegypti*, high numbers of *Ae. albopictus* could even be regarded as a factor supporting a reduced and weak transmission cycle, provided that there is no introduction of the ECSA E1-A226V-mutation CHIKV.

In terms of climate, both the Cook Islands and Vanuatu offer favorable conditions for vectors: Throughout the year maximum and minimum monthly means of air temperature as well as monthly means of precipitation are close to the hypothesized CHIKV transmissibility optimum of 25°C and 206mm respectively [45] (Table 1). We therefore conclude that neither temperature differences nor varying precipitation can be identified as crucial factors to the disparate seroprevalence levels. Population densities on the other hand have been shown to be concordant to detected seroprevalence levels (Table 1).

As CHIKV has demonstrated its potential to spread via airline routes [21,46,47,48,49,50] we further examined differences in the two island states' migration and tourism profiles: In general, Pacific islands communities share close social and economic connections, both within individual countries and internationally including high bidirectional population flows and millions of tourists visiting each year [12,15]. There are, however, distinct variations in tourism and country specific mobility patterns between the different subregions: Polynesian islands show strong individual mobility and high international migration rates, while in many Melanesian and Micronesian Island states, population movements arise predominantly within the country and international migration is at a low level [15]. Data concerning international arrivals by air (the main travel route) highlight the much higher number of individual international travels and arriving tourists per resident on the Cook Islands compared to Vanuatu (Table 1). Considering this difference, the Cook Islands CHIKV outbreak in 2014/2015, following a series of epidemics in the region, is not surprising and it can be assumed that the virus' spread to the outer island Aitutaki was facilitated by intensive (touristic) travel activities to and from the main island Rarotonga [51]. On the other hand, lower rates of residents' mobility and

tourism in Vanuatu probably contributed to the limited seropositivity detected in our survey despite an estimated high risk of virus importation [21].

To summarize, our study highlights the impact of the 2014/2015 CHIKV epidemic on the Cook Islands population and shows that a notable part of the Vanuatu test population had previous CHIKV exposure although no outbreak was reported. Comparing the different study settings, our results underline the effect of environmental and social factors on CHIKV dissemination in Pacific Islands settings and four factors potentially reducing Vanuatu's epidemic risk within the scope of CHIKV dissemination throughout the SPR were identified. These factors include presence of invasive Ae. albopictus (if no ECSA E1-A226V-mutation CHIKV is introduced), low human population densities, low international travel activity of the local population, and moderate tourism.

As with many seroprevalence surveys, limitations result from the study design [2]: Representativeness is lowered by using serum samples collected from hospital patients, rather than from the general population. Considering the comparative analysis of different seroprevalence levels we had to exclude the age group 0–9 years from standardization and extrapolation due to low case numbers. In addition, we want to emphasize that not all islands of the two island states could be included in the analysis. Regarding the isolated character of our study settings, this could lead to false estimations concerning the seroprevalence levels on islands not depicted in this survey and extrapolated seropositivity rates should be interpreted with caution.

Another limiting factor which cannot be ruled out is the risk of cross-reactions of antibodies against other alphaviruses, e.g. the Ross River Virus (RRV). RRV emerged in the SPR in 1979–1980, causing large outbreaks and there is evidence suggesting silent circulation in several PICTs ever since [52,53]. However, this hypothesis seems rather unlikely: In addition to the fact that there was no known RRV epidemic in Vanuatu in the past, the remaining seroprevalence levels would be expected to be higher in age groups already born during the epidemic (40+ years) and to rise over time in age-dependent manner in the case of silent circulation. Further, information concerning the role of Ae. hebrideus in CHIKV transmission are lacking.

In general, seroprevalence surveys are an indispensable tool for exact estimations of arboviruses' health impact, geographical distribution and trends in transmission [2]. Our findings supplement and correct the knowledge concerning CHIKV epidemics in the SPR which is often based on incomplete surveillance data. In addition, results may support preventive and rapid response measures in affected areas and are of international interest for travel-related risk assessment and infection identification in the era of globalization. However, regarding the special geographic features of the SPR, seroprevalence surveys can provide only selective information and further studies are needed to complement our knowledge concerning the real impact of emerging arboviral diseases in this region.

## Supporting information

**S1 Appendix. Test population and ELISA test results_CHIKV.**
(XLSX)

## Acknowledgments

Our special thanks go to Dominik Mildt for technical editing and continuous support as well as to Caitlin Douglas for proof-reading. We further thank DHL Cook Islands and DHL Vanuatu for logistical assistance.

## Author Contributions

**Conceptualization:** Charlotte E. B. Saretzki, Gerhard Dobler, Elisabeth Iro, Thomas Küpper.

**Formal analysis:** Charlotte E. B. Saretzki, Nicole Heussen.

**Funding acquisition:** Charlotte E. B. Saretzki, Thomas Küpper.

**Investigation:** Charlotte E. B. Saretzki, Gerhard Dobler, Yin May, Douglas Tou, Eteta Lockington, Michael Ala.

**Methodology:** Charlotte E. B. Saretzki, Gerhard Dobler, Thomas Küpper.

**Project administration:** Charlotte E. B. Saretzki, Elisabeth Iro, Thomas Küpper.

**Resources:** Gerhard Dobler, Elisabeth Iro, Yin May, Douglas Tou, Eteta Lockington, Michael Ala, Thomas Küpper.

**Supervision:** Gerhard Dobler, Thomas Küpper.

**Visualization:** Charlotte E. B. Saretzki.

**Writing – original draft:** Charlotte E. B. Saretzki.

**Writing – review & editing:** Gerhard Dobler, Nicole Heussen, Bruno S. J. Phiri, Thomas Küpper.

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
