## [Decision Letter · Decision Letter 0]

4 Aug 2022

Dear Prof. Dr. Kuepper,

Thank you very much for submitting your manuscript "Chikungunya virus (CHIKV) seroprevalence in the local population of the Cook Islands and Vanuatu and associated environmental and social factors A first CHIKV seroprevalence study in two South Pacific communities" for consideration at PLOS Neglected Tropical Diseases. As with all papers reviewed by the journal, your manuscript was reviewed by members of the editorial board and by several independent reviewers. In light of the reviews (below this email), we would like to invite the resubmission of a significantly-revised version that takes into account the reviewers' comments. 

The reviewers have made a number of suggestions about how to clarify methods that we encourage you to consider in your revision.

We cannot make any decision about publication until we have seen the revised manuscript and your response to the reviewers' comments. Your revised manuscript is also likely to be sent to reviewers for further evaluation.

Sincerely,

Gregory Gromowski

Academic Editor

Elizabeth Carlton

Section Editor

Reviewer's Responses to Questions

**Key Review Criteria Required for Acceptance?**

**Methods**

-Are the objectives of the study clearly articulated with a clear testable hypothesis stated?

-Is the study design appropriate to address the stated objectives?

-Is the population clearly described and appropriate for the hypothesis being tested?

-Is the sample size sufficient to ensure adequate power to address the hypothesis being tested?

-Were correct statistical analysis used to support conclusions?

-Are there concerns about ethical or regulatory requirements being met?

Reviewer #1: The objectives are pretty well described, however I would suggest as per PLOSNTD 'suggestions' to:

a) relocate sentences 105-109 to Methods section,

b) amend formulation > To assess the seroprevalence of CHIKV a study has been conducted in the local / (or [apparently not as expatriates were not excluded]? — age limitations?) population of the Cook Islands, which had been affected by an outbreak in 2014/15 and Vanuatu, which had never reported such an outbreak. In both study setting specific environmental variables were assessed ... (specify why?).

c) consider to add a sentence on why you selected these two islands except for the yes/no reason. There would have been many other options.

With such additional information the objective will be crystal-clear, the population clearly described. The sample size is sufficient for the simple statistical analysis needed here, except that on Vanuatu the sample size is too small apparently lacking cases in small children to show that seroprevalence has recently been acquired. No concerns about ethics or regulatory issues.

DETAILS:

114-166 I presume there is only one hospital lab each in Espiritu Santu, Rarotonga and Aitutaki. If so be more precise, if you selected single labs, explain why.

119 Do not start sentence with a number or write 'Fourhundred...'.

Reviewer #2: objectives clear, study design appropriate. population clearly indicated, sample size sufficient, statistical methods ok, no ethical concerns

**Results**

-Does the analysis presented match the analysis plan?

-Are the results clearly and completely presented?

-Are the figures (Tables, Images) of sufficient quality for clarity?

Reviewer #1: Essentially well described. Particularly appreciated that the results in the text and figures / tables were complimentary and not repetitive.

DETAIL: 

206 Why was age group 0-9y excluded? No data? Would have been relevant, see above.

213 25.0?

220-226 As the Cook Island outbreak has not been part of the study per se I would rather move that to the Discussion (or Introduction).

Reviewer #2: Analysis matches the plan, results clear, tables to be slightly modified (as indicated)

**Conclusions**

-Are the conclusions supported by the data presented?

-Are the limitations of analysis clearly described?

-Do the authors discuss how these data can be helpful to advance our understanding of the topic under study?

-Is public health relevance addressed?

Reviewer #1: Essentially agree with the conclusions presented from 293. Suggest to add the word 'international' on line 302 to more clearly differ from local mobility.

Limitations are clearly described, possibly an addition should be made relating to the pediatric population.

Suggest to add a final conclusive sentence on the public health relevance and the need of further assessment in the SPR.

Reviewer #2: conclusions clear. limitations correctly indicated, contibutions indicated, PH relevance addressed

**Editorial and Data Presentation Modifications?**

Reviewer #1: Suggest to modify the title: Chikungunya virus seroprevalence in the South Pacific populations of the Cook Islands and Vanuatu with associated environmental and social factors. > delete the second sentence which does not offer much additional information. 

While the English is pretty good, sometimes the wording is a bit clumsy (possibly from German background) > suggest to have a native British or American speaker with epidemiological knowledge check the manuscript.

PLOS NTD does not publish any titles like 'full professor', 'MD' > delete. Anyhow you have been inconsistent. See instructions for authors.

59. Suggest 'Arboviral' instead of 'mosquito-borne viral'

95. Are subregions essential for this paper? Rather not. 

98. Are natural disasters essential for this seroprevalence study?

Suggest you review the manuscript only keeping the essential elements.

50, 107, 293: no need to perseverate on 'first study' in these two islands. After all there was a seroprevalence of about 3% in French Polynesia possibly before 2014.

DISCUSSION:

228-229 Initial sentence rather trivial > suggest to delete and start: Subsequent to the 2014/15 CHIKV outbreak, 30% showed evidence ...

231 Suggest: ... differences between the two Cook Islands [the names will be specified in 234]

Reviewer #2: COMMENTS IN DETAIL:

ABSTRACT: 

BACKGROUND: 

P(age) 3 / L(ine) 35: to my knowledge there is no local transmission in Australia as by today

AUTHOR SUMMARY

P4 / L60: this is not really true; e.g. WNV circulates in the USA and in Europe too; maybe better: „as most of them predominantly circulate in (sub-)tropical regions…“

INTRODUCTION

P 5 / L 78: „global burden of disease“ instead of „global disease burden“

… / L79: are yellow fever virus, japanese encephalitis virus, west nile virus, tick borne encephalitis virus not important ? Maybe better: amongst all arboviruses the CHIKV adds …

… / L86: well it was not the extent only but also the speed (e.g. Dominican Republic: roughly 540.000 cases within 15 months) which might also be due to the aggressive biting behaviour of Aedes ssp. in contrast e.g. to Culex ssp.

… / L88: [6,7,8] instead of [6],[7],[8]

… / L90: [8,9,10] instead of [8],[9],[10]

P 6 / L 102: [9,10,13,14,15,16,17] instead of [9],[10],[13],[14],[15],[16],[17]

 „local burden of disease“ instead of „local disease burden“

… / L 104: [9,10] instead of [9],[10]

MATERIAL AND METHODS

P 7 / L 114: I wonder whether it would be better to place the comment on the votum by the ethic committee (L 124-127) at the beginning of this chapter

RESULTS

P 10 / Table: you indicate „warmest month“ and „coldest month“ with the temperature in brackets and in the colums you indicate the temperature with the month in brackets and with the precipitation you indicate the mm in brackets whereas you show the month in brackets which is not even indicated in the first column; if you use […] for description oft he value you could indicate : temperature [oC] (warmest month). Temperature [oC] (coldest month), highest precipitation [mm] (month), lowest precipitation [mm] (month)

misspelling in Ae. hebrideus

P12 / L190: „specimens“ instead of „specimen“ (..L191 idem)

DISCUSSION

P15 / L240: „real burden of disease“ instead of „real disease burden“

…/L 241: [2,13] instead of [2],[13]

P16 / L261: [32,40,41] instead of [32],[40],[41]

…/L277: [21,46,47,48,49,50] instead of [21],[46],[47],[48],[49],[50]

P17 / L280: [12,15] instead of [12],[15]

P18 / L316: [52,53] instead of [52],[53]

REFERENCES

 P23 / L 419: „cyclone“ instead of „cyclon“

**Summary and General Comments**

Reviewer #1: Well conducted study which adds to the perception of CHIKV epidemiology in the South Pacific Region.

Reviewer #2: THIS IS AN IMPORTANT STUDY THAT NOT ONLY SERVES THE LOCAL POPULATION IN TERMS OF INDIVIDUAL AS WELL AS PUBLIC HEALTH, I.E. DISEASE PREVENTION, IDENTIFICATION / DETECTION / AWARENESS AND POSSIBLE TREATMENT BUT ALSO THE TRAVELING POPULATION IN ALL THE FIELDS MENTIONNED. AS THIS IS THE FIRST STUDY ON CHIK PREVALENCE IN THIS REGION I STRONGLY SUPPORT THIS PUBLICATION AFTER MINOR REVISIONS ARE DONE

PLOS authors have the option to publish the peer review history of their article (what does this mean?). If published, this will include your full peer review and any attached files.

Reviewer #1: No

Reviewer #2: Yes: Martin Haditsch, M.D., Ph. D.
---

## [Decision Letter · Decision Letter 1]

14 Nov 2022

Dear Prof. Dr. Kuepper,

We are pleased to inform you that your manuscript 'Chikungunya virus (CHIKV) seroprevalence in the South Pacific populations of the Cook Islands and Vanuatu with associated environmental and social factors' has been provisionally accepted for publication in PLOS Neglected Tropical Diseases.

Additionally, we do suggest a minor edit to the abstract to improve clarity. Currently, it states "Comparing the different study settings, four potential outbreak-protective factors were identified: presence of Ae. albopictus (in absence of ECSA E1-A226V-mutation CHIKV), as well as** low levels of local population densities**, residents’ travel activity and tourism." It is suggested that you clarify that you are referring to human population density, not vector population density. Perhaps "low levels of human population density."

Best regards,

Gregory Gromowski

Academic Editor

Elizabeth Carlton

Section Editor

Reviewer's Responses to Questions

**Key Review Criteria Required for Acceptance?**

**Methods**

-Are the objectives of the study clearly articulated with a clear testable hypothesis stated?

-Is the study design appropriate to address the stated objectives?

-Is the population clearly described and appropriate for the hypothesis being tested?

-Is the sample size sufficient to ensure adequate power to address the hypothesis being tested?

-Were correct statistical analysis used to support conclusions?

-Are there concerns about ethical or regulatory requirements being met?

Reviewer #1: Reviewed as suggested, no further comments.

**Results**

-Does the analysis presented match the analysis plan?

-Are the results clearly and completely presented?

-Are the figures (Tables, Images) of sufficient quality for clarity?

Reviewer #1: Reviewed as suggested, no further comments.

**Conclusions**

-Are the conclusions supported by the data presented?

-Are the limitations of analysis clearly described?

-Do the authors discuss how these data can be helpful to advance our understanding of the topic under study?

-Is public health relevance addressed?

Reviewer #1: Reviewed as suggested, no further comments.

**Editorial and Data Presentation Modifications?**

Reviewer #1: Reviewed as suggested, no further comments.

**Summary and General Comments**

Reviewer #1: Reviewed as suggested, no further comments.

PLOS authors have the option to publish the peer review history of their article (what does this mean?). If published, this will include your full peer review and any attached files.

Reviewer #1: **Yes: **Robert Steffen

---

## [Editor Report · Acceptance letter]

24 Nov 2022

Dear Saretzki,

We are delighted to inform you that your manuscript, "Chikungunya virus (CHIKV) seroprevalence in the South Pacific populations of the Cook Islands and Vanuatu with associated environmental and social factors," has been formally accepted for publication in PLOS Neglected Tropical Diseases.

Best regards,

Shaden Kamhawi

co-Editor-in-Chief

Paul Brindley

co-Editor-in-Chief
